# Sequential Track–Bridge Interaction Analysis of Quick-Hardening Track on Bridge Considering Interlayer Friction

**Sanghyeon Cho** [1] , **Kyoung-Chan Lee** [2,*] , **Seung Yup Jang** [3] , **Ilwha Lee** [2] **and Wonseok Chung** [1,*]

1   Department of Civil Engineering, Kyung Hee University, Yongin 17104, Korea; 0107sanghyun@gmail.com
2   Korea Railroad Research Institute, Uiwang, Gyeonggi-do 16105, Korea; iwlee@krri.re.kr
3   Department of Transportation System Engineering, Graduate School of Transportation, Korea National University of Transportation Uiwang, Gyeonggi-do 16106, Korea; syjang@ut.ac.kr
*   Correspondence: kclee@krri.re.kr (K.-C.L.); wschung@khu.ac.kr (W.C.);
    Tel.: +82-31-460-5391 (K.-C.L.); +82-31-201-2550 (W.C.)

**Abstract:** A quick-hardening track (QHT) was developed by injecting quick-hardening mortar into an existing ballast track to rapidly substitute the ballast track with a slab track, thereby improving maintainability and running safety. QHT tracks on a bridge undergo track–bridge interactions similar to other track systems. This paper presents a model to analyze the interaction between the QHT and the bridge. This model considers the longitudinal resistances of rail fasteners and anchors, as well as the interlayer friction between the track and the bridge. A sequential analysis method was applied to systematically consider such effects, revealing that rail additional stress will be high if the track slips over the bridge for a very low frictional coefficient of 0.1. Furthermore, a track segment without an anchor can slip under train traction load when the frictional coefficient is 0.3 or lower. For low friction cases, low-speed operation is advised to prevent the accumulation of the resulting longitudinal slip displacements of the track. An anchor should be installed immediately after the quick-hardening mortar provides sufficient bearing strength to the anchors. The proposed sequential analysis is useful for determining the critical friction coefficient and appropriate longitudinal resistance of a rail fastener, as well as for verifying track safety.

**Keywords:** quick-hardening track; railway bridge; track–bridge interaction; sequential interaction analysis; friction

## 1. Introduction

Ballast tracks are inexpensive to construct and lines can be easily altered if needed; however, track irregularity is generated due to gradual plastic deformation in the ballast layer during operation. The lifetime of a ballast track is reduced when the train load is repeatedly applied [1], and track irregularity progresses owing to ballast degradation and water permeation [2]. A similar irregularity can be observed at the contact wire with the interaction of a pantograph [3]. The expected lifetime of ballast track is approximately 30 years even with regular maintenance [4]. The maintenance cost of railway tracks increases particularly for a high-speed line due to excessive vibration and subsidence [5].

Slab tracks were developed to prevent track irregularity and to reduce the maintenance cost. However, it is only being applied to newly constructed lines since rapid construction is not feasible due to the time required for concrete curing. A quick-hardening track (QHT) was developed as a method for rapidly substituting deteriorated ballast tracks with slab tracks without needing train operations to be stopped [6–12]. A QHT is constructed by first collecting the ballast from the existing track and then

washing and laying the washed ballast back in a mold made of geo-textile, and finally injecting the quick-hardening mortar.

When installing a slab track including a QHT on railway bridges, the longitudinal displacement of tracks may occur due to the thermal expansion and contraction of the bridge, as well as the traction and braking loads of the train. To prevent such displacement, a shear key structure should be installed between the bridge deck and the track to rigidly connect the track slab and bridges. A QHT on a bridge deploys such a structural connection as an anchor block with shear studs and post-installed anchors between the bridge deck and the track slab.

The QHT slab is vulnerable to cracks, as reinforcement is not laid within the track slab. To avoid restrained stress and the resulting cracks, the QHT on a bridge has disconnected segments at 5-m intervals to avoid confining the temperature deformation of the track slab, and the anchor block is installed only at the central part of each segment. This anchor block should be installed after the poured mortar develops sufficient strength to prevent the early damage of the track slab in the vicinity of the anchor block [13]. Without the anchor, the longitudinal resistance between the QHT and the bridge only comes from the frictional forces between the underside of the track and the top of the bridge deck. After the anchor block is installed, this resistance comes from the shear resistance of the anchors and surface friction. Therefore, both anchor and friction must be taken into consideration for the track–bridge interaction of QHTs on bridges.

The additional axial stresses of the continuous welded rail on a bridge increases as the bridge span lengthens due to the track–bridge interaction. According to the related standards [14,15], including UIC 774-3R [16], the longitudinal track–bridge interaction should be analyzed to examine whether the additional rail axial stress of the continuous welded rail exceeds the allowable stress [17–19]. Previous studies [20,21] have presented the track–bridge interaction of QHTs on railway bridges. The appropriate installation time for anchors and the possible normal operation time of trains were examined after building QHTs with respect to various frictional coefficients between the track slab and the bridge deck [13]. The longitudinal and transverse structural behavior of QHTs on bridges were examined through an experiment program using actual-size QHT specimens; subsequently, the minimum required frictional coefficient was deduced [22,23].

There are two methods for deducing track–bridge interactions. First, a separate analysis method simply sums the stress and deformation generated by the temperature and train load, respectively. Second, a sequential analysis method solves the stress and deformation in a sequential manner in the order of temperature and train-load generation. Sequential analysis faithfully examines the behavior of the track and the bridge; however, its implementation is rather complicated. It is known that separate analysis conservatively evaluates rail stress compared to sequential analysis, whereas it is the other way around with regard to displacement [24]. A separate analysis is commonly conducted in field projects, as investigated in previous studies [19–23]. For implementing sequential analysis, the structural properties of the longitudinal resistance of the track should consider the existence of train load after temperature changes [25,26]. As the longitudinal resistance normally comes from a rail fastener in a typical slab track, the typical sequential analysis considers nonlinear behavior between the rail and track without the frictional behavior between the track and the bridge.

QHT track segments and the bridge are structurally connected by anchor blocks that comprise a steel bracket, laterally welded stud shear connectors, and post-installed anchors. Figure 1 shows the construction order of QHTs on railway bridges [22]. After removing all ballasts on the existing ballast track, anchor brackets are positioned at the center of each segment of the QHT (Figure 1a). The washed ballasts are laid on a cast form using geotextiles on top of the bridge deck (Figure 1b). The track skeletons, to which concrete sleepers and rails are connected, are installed on the laid ballast (Figure 1c). Once the laid ballast stabilizes after a few days, quick-hardening mortar is injected and cured (Figure 1d). When the quick-hardening mortar exhibits sufficient strength, the post-installed anchors are installed on the bridge deck (Figure 1e).

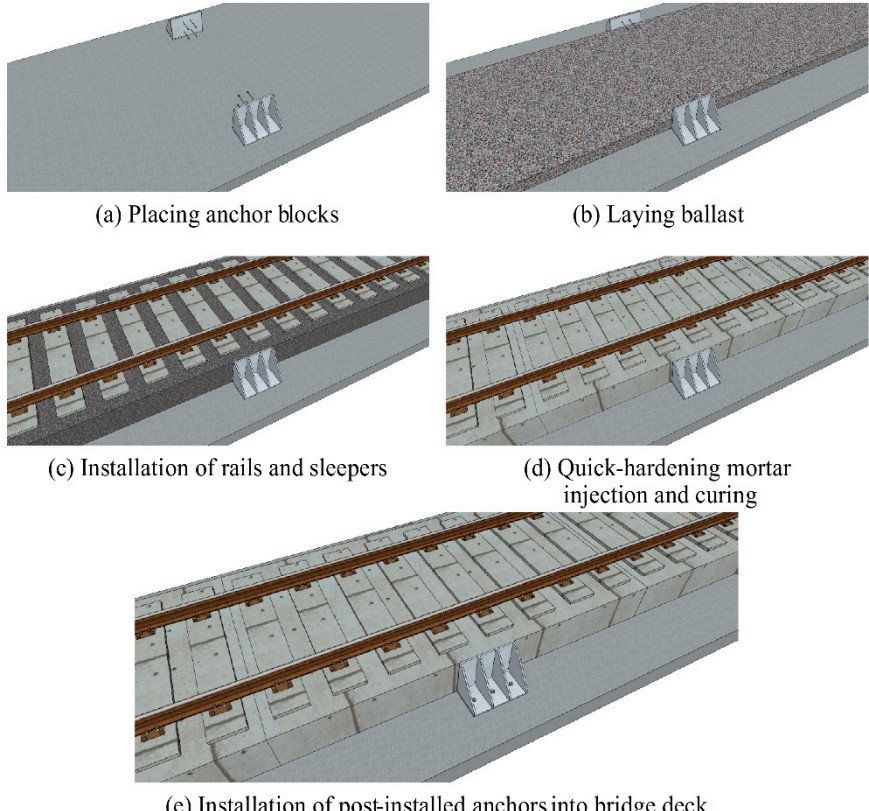

(a) Placing anchor blocks

(b) Laying ballast

(c) Installation of rails and sleepers

(d) Quick-hardening mortar injection and curing

(e) Installation of post-installed anchors into bridge deck

**Figure 1.** Concept of the anchor system for a quick-hardening track [22].

After the post-installed anchors are installed, the shear resistance of the anchor blocks can withstand temperature and train loads. If a longitudinal load is delivered to the anchor blocks before the injected quick-hardening mortar gains sufficient strength and hardens, cracks may occur in the quick-hardened concrete where the shear connectors are embedded. Thus, the post-installed anchors must be installed after the mortar develops sufficient strength. The compressive strength of quick-hardening mortar is 9.6 MPa two hours after curing, and 30 MPa after one day of curing [27]. Therefore, the post-installed anchors can be installed when the QHT is at least one day old. Before installing post-installed anchors, the longitudinal resistance only comes from the frictional forces between the underside of the QHT and the top of the bridge deck.

In this study, the track–bridge interaction is examined when only frictional resistance exists before the post-installed anchors are installed. Here, examining the longitudinal relative displacement of QHT segments, as well as rail stresses due to the interaction, are important. A sequential analysis was performed herein because a separate analysis may underestimate the displacement. The friction between track and bridge and the structural behavior of an anchor were evaluated based on the experimental results of a previous study [22], and the effects caused by a change in friction coefficient were also examined.

## 2. Track–Bridge Interaction Analysis Model

The analyzed bridge is a simple one, having a span of 40.96 m, whereas the length, width, and height of the track slab segment are 5.12 m, 2.8 m, and 0.53 m, respectively. Eight track slab segments were installed on the bridge. An embankment of 31 m on both sides of the bridge was taken into consideration. The sectional constant and material properties of the bridge, track, and rail are shown in Table 1.

**Table 1.** Material and sectional properties of rail and PSC (pre-stressed concrete) box girder.

| Case | Equivalent Elastic Modulus (MPa) | Section Area (mm$^2$) | Span Length (m) | 2nd Moments of Inertia (mm$^4$) |
|---|---|---|---|---|
| Rail (60E1) | 210,000 | $1.53 \times 10^4$ | 100 | $6.07 \times 10^7$ |
| QHT slab | 28,571 | $1.48 \times 10^6$ | 5.12 | $3.47 \times 10^{10}$ |
| PSC box girder | 28,571 | $1.19 \times 10^8$ | 40.96 | $12.12 \times 10^{13}$ |

The analysis of the track–bridge interaction was conducted using a commercial structural analysis program, ABAQUS [28]. The continuous welded rail, track slab segment, and bridge were modeled using planar Timoshenko beam elements (B21), as shown in Figure 2.

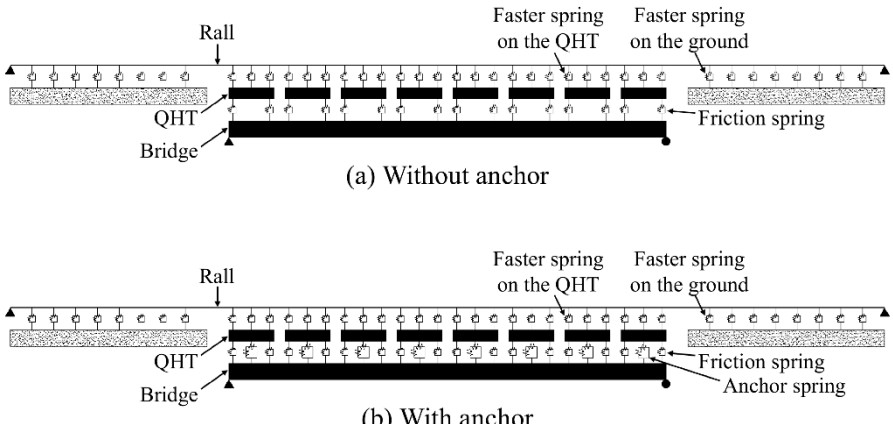

(a) Without anchor

(b) With anchor

**Figure 2.** Schematic of rail–track–bridge interaction model for a QHT (quick-hardening track).

As rails are connected by rail fasteners installed on the sleepers embedded in the track slab, rails and track slabs are connected to each other using a spring element that replicates the longitudinal resistance of rail fasteners (hereinafter referred to as the "fastener spring"). The longitudinal resistance by a fastener is replicated using a bilinear spring that exhibits elastic-perfect plastic behavior, as shown in Figure 3. The displacement that sets the boundary between the elastic and plastic zone is defined as the limit displacement ($u_0$). Until the displacement is less than the limit displacement, the rail pad itself deforms due to elastic shearing. After the limit displacement is exceeded, the rail slips over the rail pad, and the longitudinal displacement increases between the rail and the track slab. When there is no train on the track, the longitudinal resistance is determined by the normal force due to the clamping force of a fastener and the friction coefficient of a rail pad. When there is a train on the track, the longitudinal resistance is determined by the vertical load of the train and the friction coefficient of a rail pad. Therefore, the maximum longitudinal resistance of a fastener spring is different when there is a train load (60 kN/m) and when there is no train load (40 kN/m), as suggested in UIC 774-3R [16]. The limit displacement is generally as small as 0.5 mm in the same code.



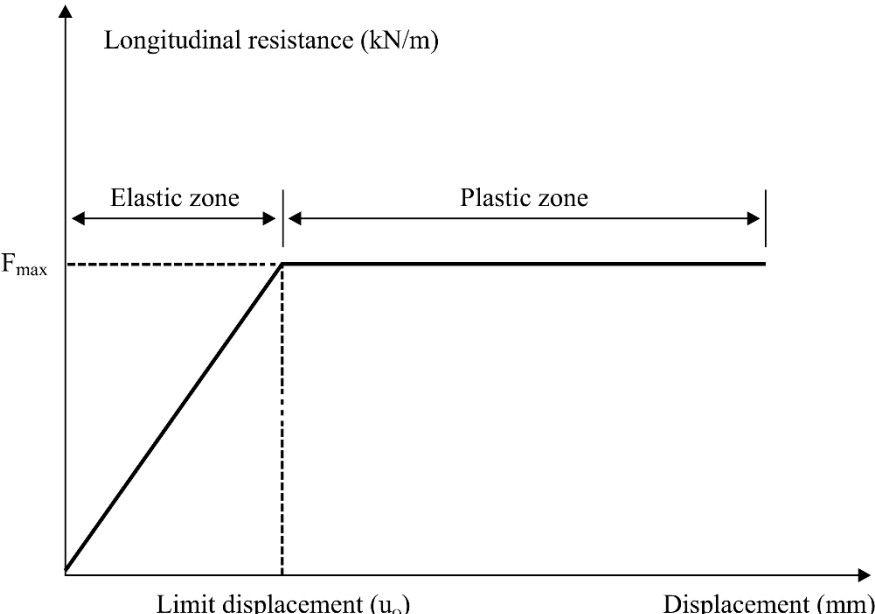

**Figure 3.** Definition of a bilinear spring of a rail fastener and friction longitudinal resistance.

In QHT, considering the frictional resistance between the track slab and the railway bridge is important [13,22,23]. The track slab and the bridge are connected by a nonlinear spring element that replicates the frictional behavior (hereinafter, the "friction spring"). When there is an anchor installed, a spring element replicating the resistance due to an anchor block is added to the center of the track segments (Figure 2). The longitudinal resistance due to the friction between track and bridge has a pattern similar to the longitudinal resistance of a fastener. When there is no train on the track, the longitudinal resistance is developed due to the normal force from the self-weight of the track segment and the frictional coefficient between the track slab and the bridge deck.

In this study, considering that one QHT track slab has a volume of 7.6 $m^3$, a unit weight of 24 $kN/m^3$, and a friction coefficient of 0.7, the maximum kinetic frictional force of a track segment is 127.65 kN without the train. This corresponds to the maximum longitudinal resistance of a friction spring of 24.9 kN/m (=127.65 kN/5.12 m), considering a segment length of 5.12 m. When there is a train on the track, the longitudinal resistance is obtained by summing the track self-weight load and train load of 80 kN/m.

The limit displacement of a friction spring was applied with 0.05 mm, obtained from the experiment program in a previous study [22]. The friction coefficient in the test was 0.7; however, it may change at actual field sites because the geo-textile in the friction layer can deteriorate, become soaked, or freeze. To examine the effects of a change in the friction coefficient, analyses were conducted by decreasing the friction coefficient from 0.7 to 0.1 by intervals of 0.2. Table 2 presents the limit displacement and the maximum resistance of bilinear springs that replicates the longitudinal resistance of a rail fastener and the frictional layer between the track and bridge with or without the train load.

**Table 2.** Bilinear properties of fastener and friction springs.

| Case | Friction Coefficient | Limit Displacement, $u_o$ (mm) | Maximum Longitudinal Resistance, $F_{max}$ (kN/m) Without Train Load | Maximum Longitudinal Resistance, $F_{max}$ (kN/m) With Train Load |
|---|---|---|---|---|
| Rail fastener spring | - | 0.5 | 40.0 | 60.0 |
| Friction spring | 0.7 | 0.05 | 24.9 | 80.9 |
| | 0.5 | 0.05 | 17.8 | 57.8 |
| | 0.3 | 0.05 | 10.7 | 34.7 |
| | 0.1 | 0.05 | 3.6 | 11.6 |

## 3. Sequential Interaction Analysis Scheme

A separate analysis, which involves examining the effects of temperature and train load separately and then simply summing the results for evaluation, underestimates the resulting displacement [24]. Therefore, a separate analysis is not appropriate for evaluating the relative displacements of QHT segments due to track–bridge interactions. As shown in Figure 4, a separate analysis overestimates the longitudinal resistance when the train load is applied after the temperature load. Previous studies [13,21] used the separate analysis scheme to evaluate the safety of anchor blocks, as it handled applied forces conservatively. However, this is not the case for displacement evaluation. Before anchors are installed, evaluating the longitudinal displacement is important, and the separate analysis is not appropriate in this regard. A sequential scheme can accurately evaluate the longitudinal forces and displacements due to interaction. In this study, therefore, a sequential analysis was applied to consistently analyze the longitudinal behavior of a QHT on the railway bridge.

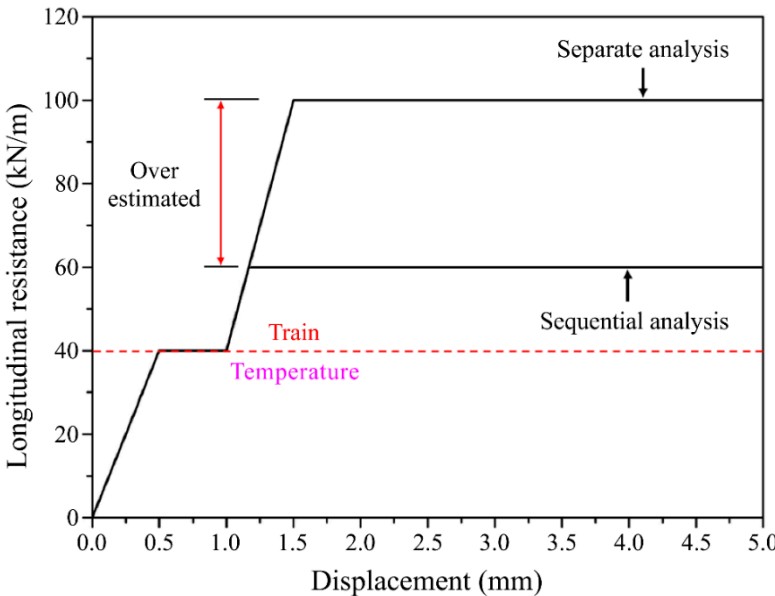

**Figure 4.** Longitudinal resistance determined through separate and sequential analyses.

Equations (1) and (2) show the corresponding maximum and minimum longitudinal resistances calculated by a sequential analysis when the loading state changes from having only a temperature load to having a train load additionally applied. The maximum and minimum longitudinal displacements generated in the elastic zone are shown in Equations (3) and (4), respectively.

$$F_{max}^{i+1} = -(F_l - F_u) + F_{max}^i - F_i \tag{1}$$

$$F_{min}^{i+1} = -(F_l - F_u) + F_{min}^i - F_i \tag{2}$$

$$u_{max}^{i+1} = \left(-\frac{1}{\alpha} + 1\right)u_0 + \left(u_{max}^i - u_i\right)/\alpha \tag{3}$$

$$u_{min}^{i+1} = \left(\frac{1}{\alpha} - 1\right)u_0 + \left(u_{min}^i - u_i\right)/\alpha \tag{4}$$

where $F_l$ is the longitudinal resistance with train load, $F_u$ is the longitudinal resistance without train-load, and $\alpha$ denotes their ratio (i.e., $F_l/F_u$).

As interaction analysis models used in previous studies [25,26] comprise the rail and bridge, spring elements were introduced only for a fastener spring. This study expands the model to the rail–track–bridge system that considers the rails, QHT, and bridge. A full sequential analysis should include both the rail fastener spring between the rail and the track and the friction spring between the track and the bridge.

Figure 5 shows the example of the nonlinear behavior of the longitudinal resistance spring. Spring type A is for when the train load is applied while the longitudinal resistance spring is within the elastic range due to the temperature load; spring type B is for when the train load is applied when a longitudinal slip has occurred and the longitudinal resistance reaches its maximum limit owing to a temperature change.

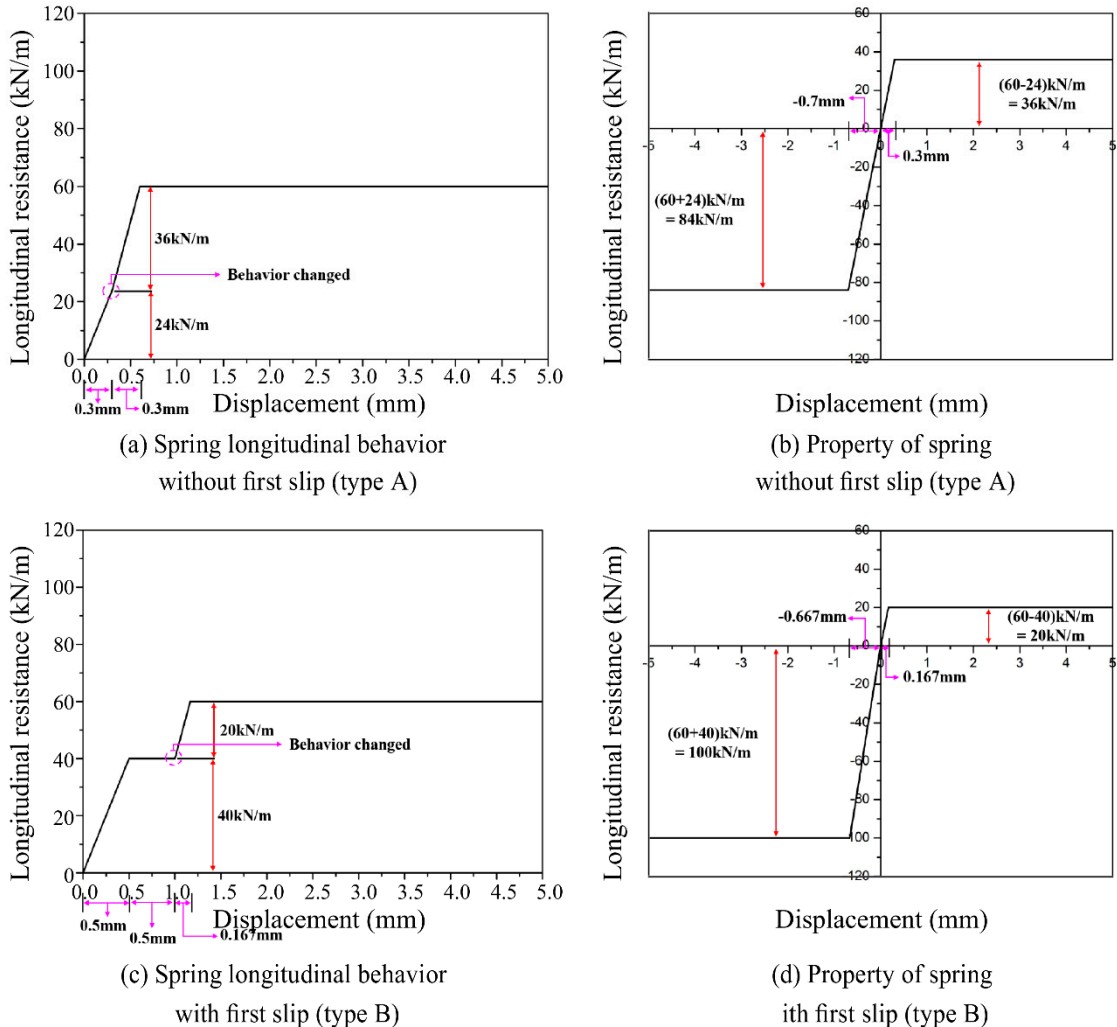

(a) Spring longitudinal behavior
without first slip (type A)

(b) Property of spring
without first slip (type A)

(c) Spring longitudinal behavior
with first slip (type B)

(d) Property of spring
ith first slip (type B)

**Figure 5.** Example of longitudinal resistance of springs obtained from sequential analysis scheme.

The example of spring type A shown in Figure 5a is when the longitudinal resistance of 24 kN/m occurs, with only the temperature load being applied without the train load. Subsequently, the maximum longitudinal resistance increases by 36 kN/m to reach 60 kN/m when the train load

is applied. Figure 5b shows the nonlinear spring computed for the train load after the longitudinal resistance of 24 kN/m is applied due to temperature load, as shown in Figure 5a. The positive resistance is reduced by the amount of resistance against the temperature load, as the temperature load was generated in the positive direction. In contrast, the negative resistance is increased by the same amount as the positive resistance. The example of spring type B shown in Figure 5c is when a slip occurs as the longitudinal resistance reaches its maximum limit of 40 kN/m due to the temperature load. Subsequently, the maximum longitudinal resistance increases by 20 kN/m when the train load is applied. Figure 5d shows the nonlinear spring properties for the train load after the longitudinal resistance reaches its maximum due to the temperature load.

This sequential analysis scheme is used for analyzing the track–bridge interaction for the target bridge (Figure 2 and Table 1). The temperature, train traction, and train vertical load are applied in compliance with the related standards [15,16], as shown in Table 3.

**Table 3.** Load definitions.

| Temperature Variation | Traction Load | Train Vertical Load |
|:---:|:---:|:---:|
| 25 °C | 33 kN/m (with L = 33 m) | 80 kN/m |

The temperature increase in the QHT slabs and bridge was 25 °C, in accordance with the standards on the temperature load of a concrete structure suggested in the Korean Design Standard [15]. The traction load of 33 kN/m, which has a greater loading effect than the braking load of a train, was applied to a loaded length of 33 m. A vertical train load of 80 kN/m was applied according to the cargo/passenger mixed load of KRL-2012 [15], which is the standard train load in Korea.

When the direction of traction load is from left to right, an increase in the additional rail axial stress has the greatest effect because the directions of the rail stress due to thermal increase and traction are identical. When the direction of traction load is from right to left, the longitudinal load of the track is the greatest on each spring because the directions of the rail deformation due to the traction load on the fastener and friction spring and the bridge deformation due to thermal increase load are opposite. Owing to such reasons, the direction of traction load was set to left-to-right when evaluating the additional rail axial stress, and to right-to-left when evaluating the longitudinal relative displacement of the track, as shown in Figure 6.

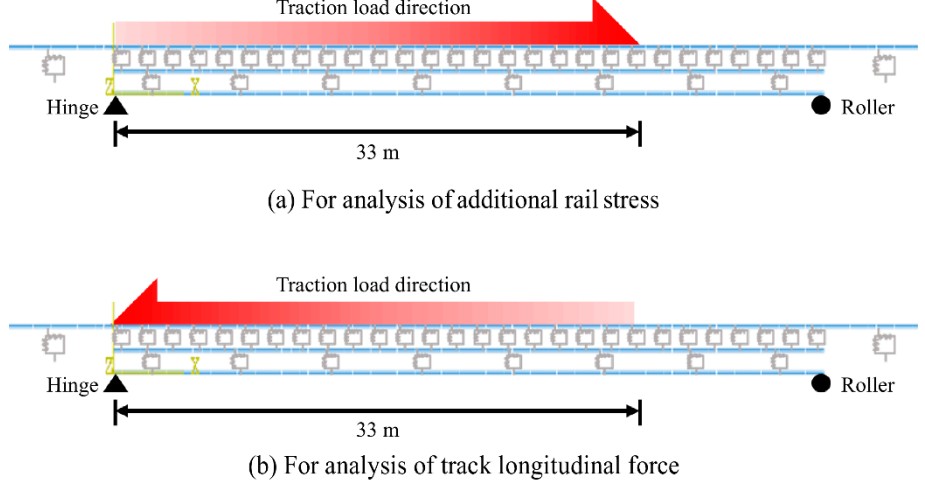

(a) For analysis of additional rail stress

(b) For analysis of track longitudinal force

**Figure 6.** Traction load direction.

## 4. Analysis Results and Discussion

### 4.1. Additional Rail Axial Stress

The additional rail axial stress of the slab track due to the track–bridge interaction should not exceed 92.0 MPa for both compression and tension [16]. A sequential analysis was performed for the additional rail axial stress due to thermal and train loads when the anchors were not installed on the QHT. Figure 7 shows the additional rail axial stress due to temperature and train loads depending on the friction coefficients between the track and the bridge.

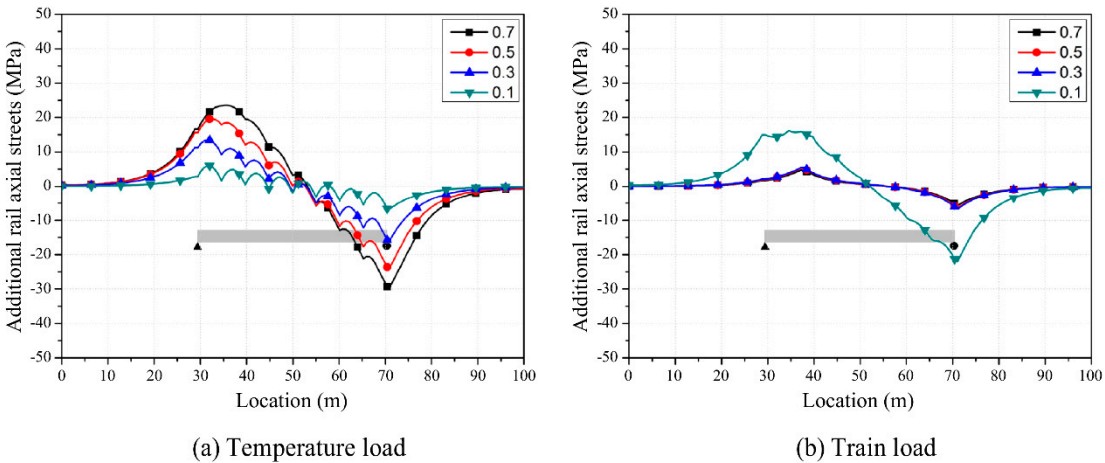

(a) Temperature load          (b) Train load

**Figure 7.** Additional rail axial stresses.

The additional rail axial stress due to temperature load increases with an increase in the friction coefficient (Figure 7a). The longitudinal bridge expansion due to a temperature change is delivered to the rail by friction and fastener springs, thereby causing additional rail axial stress. The additional axial stress on rails decreases as the longitudinal resistance from friction gets smaller, as transferred displacement from the friction layer gets smaller.

Figure 7b illustrates the additional rail stresses generated by applying the train load while considering the deformed state due to the temperature load based on a sequential analysis scheme. When the friction coefficient equaled 0.3 or higher, the same level of rail stress was generated. When the friction coefficient was 0.1, the rail stress increased because the track segment slipped from the train load. When the friction coefficient was 0.1, the stress from the train load was greater than that from the temperature load. Hence, the effect of the train load was dominant.

In a conventional analysis of track–bridge interactions, an increase in rail stress due to the traction and braking loads of a train is caused by the drift of a bridge pier. A bridge pier was not taken into account for this analysis; therefore, a change in stress due to the train load should be small because only the bending of a bridge due to train vertical load and end rotation causes such a change. However, as shown in the analysis results, a great amount of additional rail stress was generated due to traction loads when the friction coefficient of the underside of the track slab was 0.1, which is very small and unlikely to happen in practice.

If the bridge span is longer than the targeted bridge or if multiple bridges are adjacent to each other, the effect of traction loads can increase significantly, Therefore, it is recommended that the friction coefficient under the QHT should be maintained at 0.3 or higher before anchor installation. When the frictional coefficient may become excessively small, a low-speed operation should be guided so as to not induce traction/braking loads.

### 4.2. Applying Longitudinal Forces to Track Segments

The longitudinal forces applied by track slab segments were analyzed to identify the track–bridge interaction depending on the friction coefficient of QHTs on the bridge. The longitudinal forces of fasteners were calculated by summing the loads delivered from eight rail fasteners installed on one track segment slab with a length of 5.12 m. The frictional longitudinal forces were calculated by summing the loads applied on friction springs under the track slab segment.

When only the temperature load was applied (Figure 8a), the frictional longitudinal forces were smaller than the fastener longitudinal forces for all friction coefficients; thus, track segments slipped over the bridge deck. When both temperature and train loads were applied and the friction coefficient was 0.7 (Figure 8b), the frictional longitudinal forces were greater than the fastener longitudinal forces when the rail slipped over the fasteners. The maximum fastener longitudinal force, 307.2 kN, was calculated by multiplying the segment length of 5.12 m with the maximum fastener longitudinal forces of 60 kN/m under the train load. When the friction coefficient equaled 0.5 or less, the maximum frictional longitudinal forces were smaller than the maximum longitudinal forces of the rail fasteners. In this case, the track segment slipped over the bridge deck because a longitudinal load greater than the frictional resistance was applied.

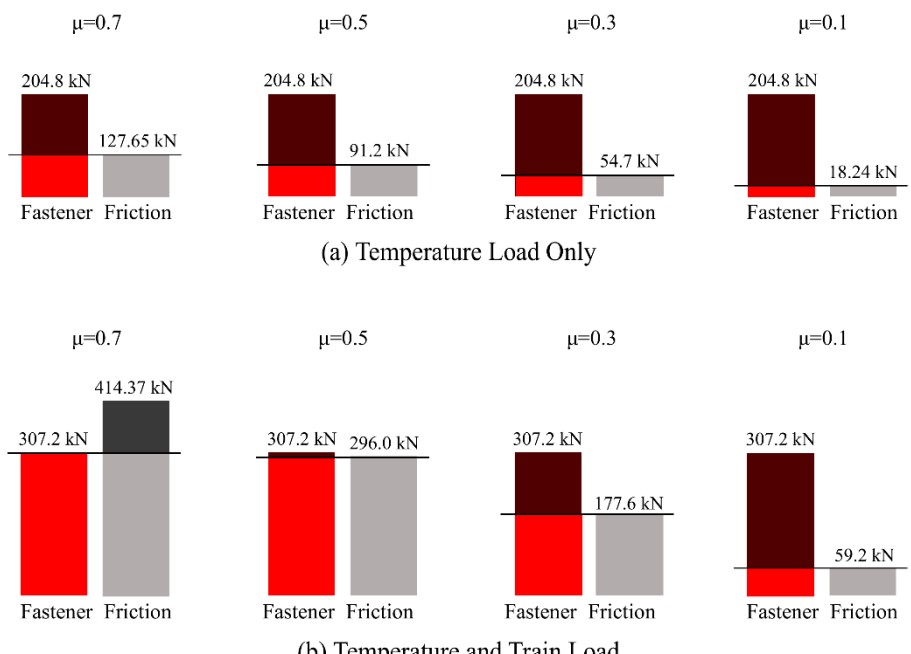

**Figure 8.** Maximum track longitudinal force with respect to friction coefficients under the QHT slab.

Figure 9 shows the applied longitudinal forces to each segment according to the friction coefficients. When the friction coefficient was 0.7 (Figure 9a), the bridge expanded in the negative direction when a temperature load was applied to the bridge and QHT, while the rail fasteners exhibited resistance in the negative direction because they resisted the movement of the bridge. The longitudinal load due to temperature load reached the maximum frictional resistance of 127.65 kN, thereby generating a track segment slip over the bridge deck. When the train load was applied, the rails were pushed in the negative direction. Fastener springs were applied with a longitudinal load in a negative direction for all sections. The maximum longitudinal load under a train load was 196.9 kN, which was less than the maximum longitudinal resistance of 307.2 kN with the train load taken into consideration, thereby not generating an additional slip.

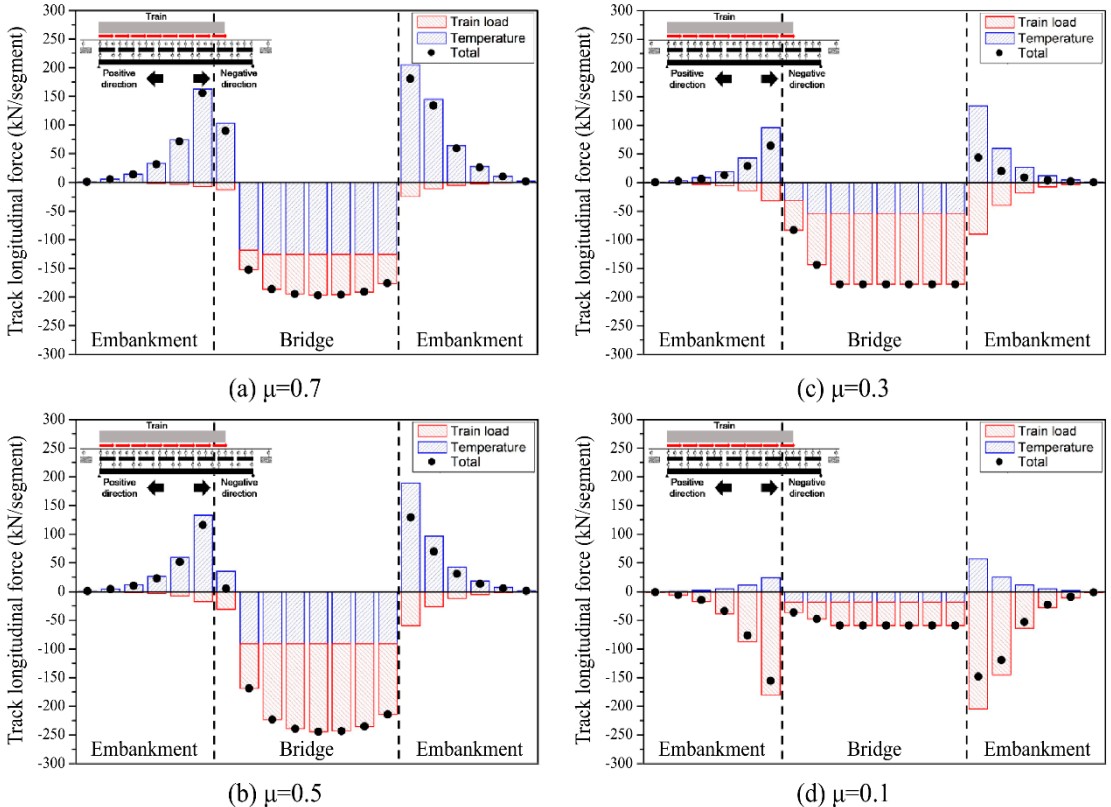

**Figure 9.** Applying longitudinal forces to the track slab segment.

When the friction coefficient was 0.5 (Figure 9b), the frictional resistance due to temperature load reached its maximum of 91.2 kN; thus, the track segments slipped. The maximum longitudinal load under a train load was 244.2 kN, which did not exceed the maximum longitudinal resistance of 296.0 kN under a train load, thereby not generating an additional slip due to train operation.

When the friction coefficient was 0.3 (Figure 9c), the maximum frictional resistance was reduced to 54.72 kN; thus, track segments slipped under the temperature loads. The maximum resistance of 177.6 kN was reached when the train load was applied; thus, the track slipped additionally owing to the train load.

When the friction coefficient was 0.1 (Figure 9d), the maximum frictional resistance was 18.24 kN, which is very small; thus, track segments slipped due to the temperature load. Even when the train load was applied, the maximum longitudinal resistance remained at 59.2 kN; thus, the track segments slipped additionally under the train load as well.

When the friction coefficient equaled 0.5 or higher, the longitudinal load of the track due to temperature and train load was smaller than the maximum frictional longitudinal resistance, thereby not generating a slip due to train load. However, when the friction coefficient equaled 0.3 or less, the maximum frictional longitudinal resistance was smaller, wherein the longitudinal load to the track exceeded the frictional longitudinal resistance, thus generating a slip due to the train load. Therefore, before the post-installed anchors are installed, the friction coefficient between the QHT slabs and bridge deck slabs should be maintained at 0.5 or higher in order to prevent a slip due to the train load. Figure 10 illustrates the relation between the applied longitudinal forces and the longitudinal resistance from fastener and friction springs with respect to frictional coefficients.

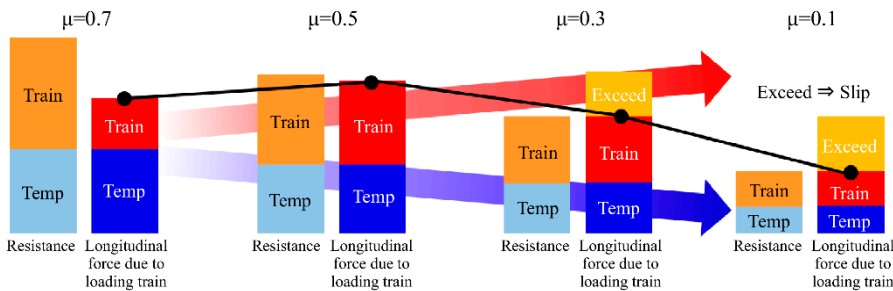

**Figure 10.** Concept of resistance–track longitudinal force relationship.

### 4.3. Relative Slips of Friction and Fastener Springs

The longitudinal displacement generated in a fastener spring is the relative displacement between the rail and QHT slabs. The longitudinal displacement generated in a friction spring is the relative displacement between the bridge deck and the QHT segment slabs. Figure 11 presents an example showing the behavior of a nonlinear resistance spring in a sequential analysis. The first slip occurs due to temperature when the maximum longitudinal resistance is reached due to the temperature load; subsequently, the secondary slip due to the train load occurs when the train load causes the maximum longitudinal resistance to be exceeded.

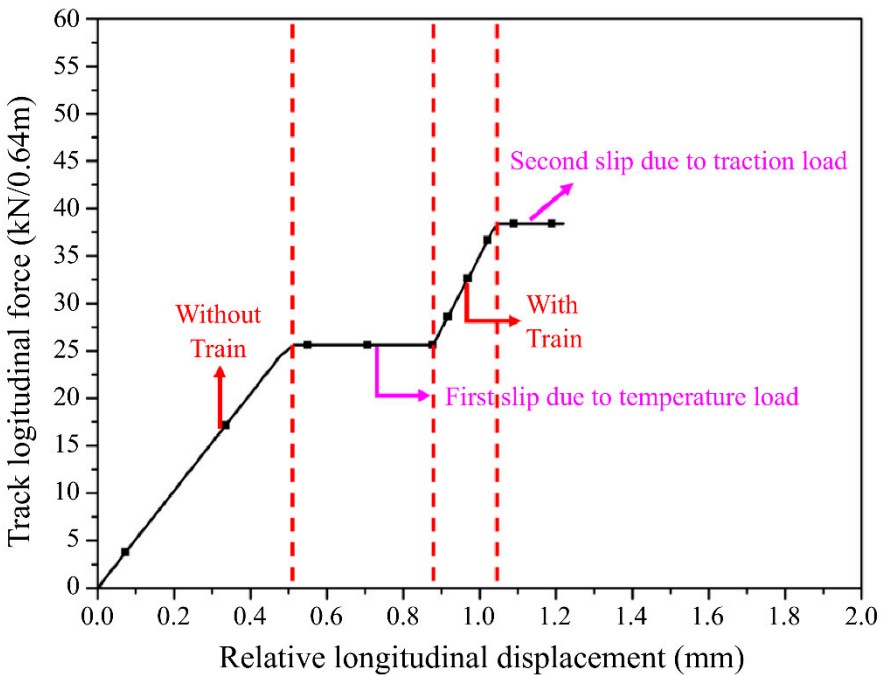

**Figure 11.** Typical sequential behavior of friction spring.

When the longitudinal relative displacement is examined, the spring with the largest first slip due to the temperature load was generated at the right-end segment, while the spring with the largest secondary slip due to the train load was generated at the central segment. Figure 12a illustrates the history of the relative displacement of a fastener spring at the right end. The secondary slip due to the train load did not occur when the friction coefficient was 0.3 or higher. When the friction coefficient was 0.1, the longitudinal displacement increased to 0.69 mm, and the train load was applied after the temperature load. Then, when the applied load at the friction spring exceeded its maximum resistance, the effect of the train load was redistributed to the embankment section; thus, the relative longitudinal displacement at the fastener spring decreased. Figure 12b illustrates the history of the relative displacement of a fastener spring located at the left end of the central segment. Rail slip due to

both temperature and train loads occurred when the friction coefficient was 0.3 or higher. In contrast, when the friction coefficient was 0.1, a rail did not slip due to the train load.

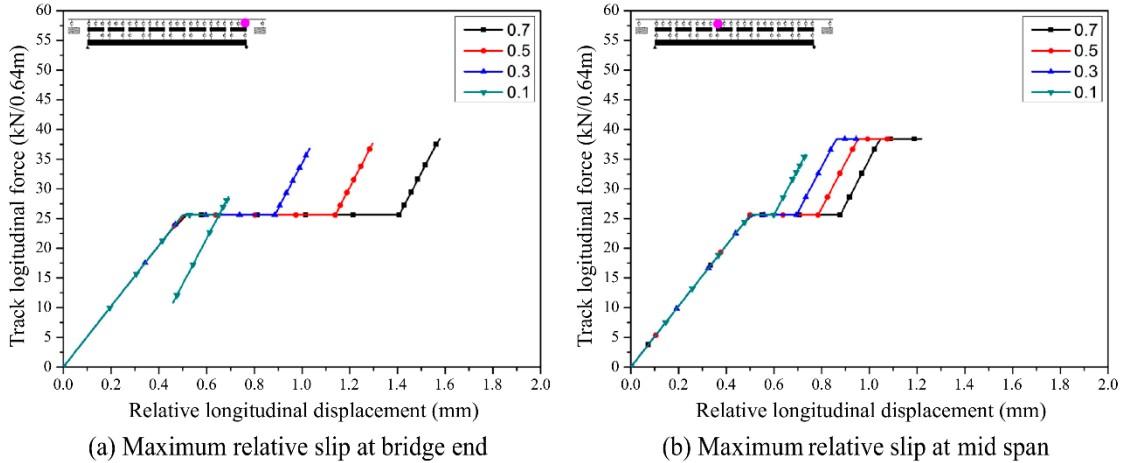

**Figure 12.** Longitudinal relative slip of a fastener spring.

Figure 13a illustrates the history of the relative displacement of a friction spring under the track segment at the right end of the bridge. The secondary slip due to the train load did not occur when the friction coefficient was 0.5 or higher, while the secondary slip occurred when the friction coefficient was 0.3. A final slip of 10.67 mm occurred when the friction coefficient was 0.1. Figure 13b illustrates the history of the relative displacement of a friction spring at the left end of the central segment. The secondary slip due to the train load did not occur when the friction coefficient was 0.5 or higher, while the slip due to both the temperature load and train load occurred when the friction coefficient was 0.3 or less.

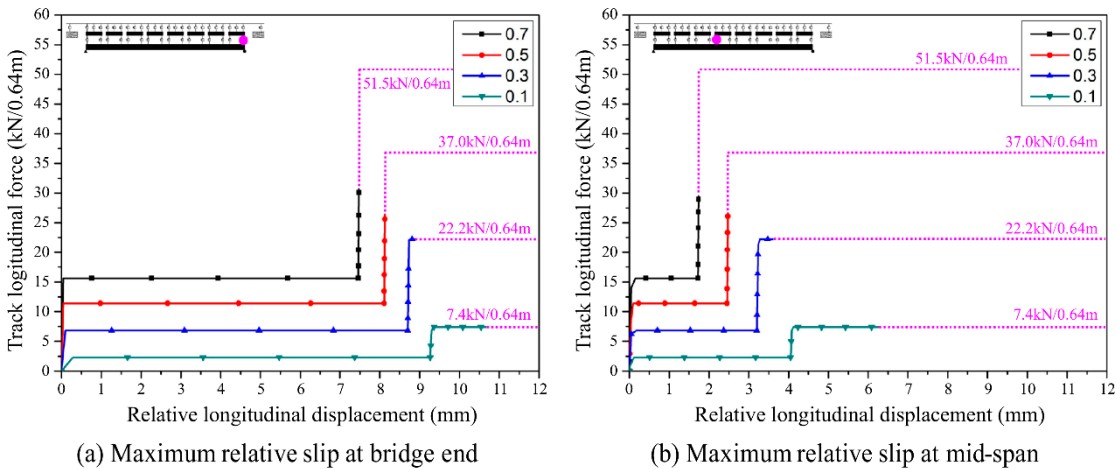

**Figure 13.** Longitudinal relative slip of a friction spring.

The maximum slip of the QHT segment occurred in the roller support at the right end of the bridge (Figure 14). Longitudinal relative displacements of 7.46 mm and 0.01 mm occurred due to the temperature and train loads, respectively, when the friction coefficient was 0.7. Longitudinal relative displacements of 9.27 mm and 1.40 mm occurred due to the temperature load and train load, respectively, when the friction coefficient was 0.1.

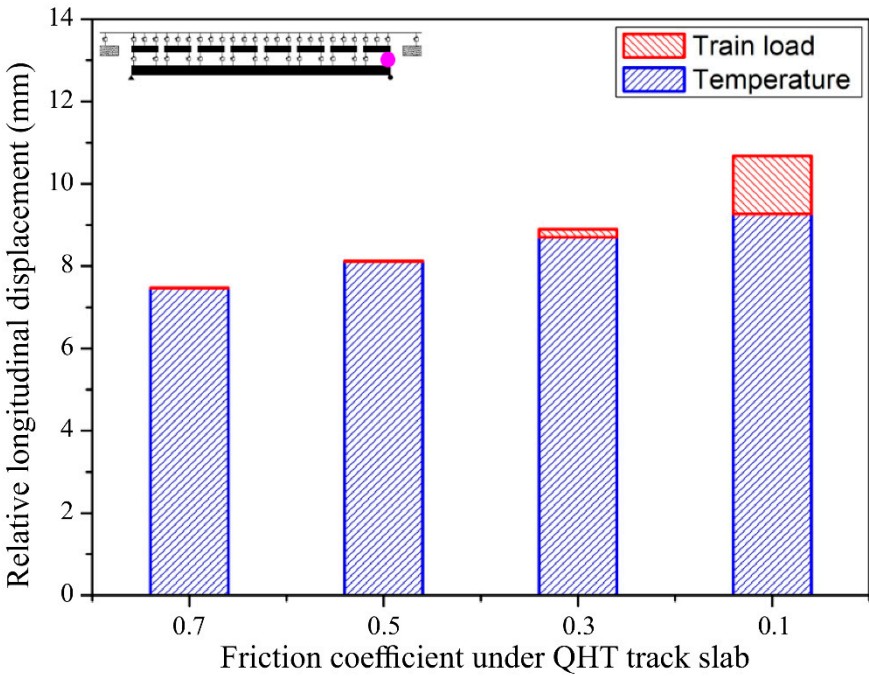

**Figure 14.** Maximum relative longitudinal displacement between a QHT slab and bridge deck.

The maximum slip due to the temperature load was less than 10 mm. The actual relative displacement in actual field applications will stay less than this, as the temperature is repeatedly altered on a daily basis. The secondary slip due to the train load was smaller in size than the first slip due to the temperature load; however, secondary slips may accumulate if trains are operated, as it continuously works in the same direction. When the friction coefficient was 0.5 or higher, only the first slip due to the temperature load occurred without the secondary slip due to the train load; hence, the longitudinal displacement from the train operation was not accumulated. When the friction coefficient was 0.3 or less, the secondary slip occurred from the train operation, which may eventually accumulate if trains are operated repeatedly, thereby making it difficult to control the longitudinal displacement.

The QHT on the railway bridge of this study was designed such that the post-installed anchors could resist the effects of train and temperature loads when the friction coefficient between track and bridge was 0.3 or higher. The shear resistance of the anchors was 240 kN according to the structural calculations. In a previous study [23], an analysis was performed on the track–bridge interaction of QHTs on which the post-installed anchors are installed. As intended in the design, the effect of the load dis not exceed the anchor resistance until the friction coefficient reached 0.3. Table 4 shows the maximum applying loads at the anchor block and the load to design resistance ratio depending on the frictional coefficient between the track and the bridge when anchors are installed.

**Table 4.** Maximum applied load at anchors with respect to friction coefficient.

| Frictional Coeff | Train | | Temperature | | Total | | Remark |
|---|---|---|---|---|---|---|---|
| | Load [kN] | Load/Resistance Ratio | Load [kN] | Load/Resistance Ratio | Load [kN] | Load/Resistance Ratio | |
| 0.7 | 5.60 | 0.023 | 26.63 | 0.111 | 32.23 | 0.134 | O.K. |
| 0.5 | 7.69 | 0.032 | 114.80 | 0.478 | 122.49 | 0.510 | O.K. |
| 0.3 | 12.35 | 0.051 | 151.40 | 0.631 | 163.75 | 0.682 | O.K. |
| 0.1 | 84.98 | 0.354 | 186.99 | 0.779 | 271.97 | 1.133 | N.G. |

As examined in the analysis, the friction coefficient between the track slab and bridge deck should be maintained at at least 0.5 before anchors are installed for QHTs on a railway bridge. When the friction coefficient is expected to be less than 0.5, trains should be advised to operate slowly to minimize the traction and braking loads. It is not easy to maintain the friction coefficient of tracks installed at

sites; therefore, anchors should be installed immediately when quick-hardening concrete develops sufficient strength after constructing QHTs.

If the bridge span arrangement, material properties, and loading conditions are different from those of the bridge structure considered in this research, the results should be analyzed accordingly using the method proposed in this paper. The resulting critical frictional coefficient required to avoid the slippage of track segments varies with these conditions.

## 5. Conclusions

In this study, an analysis model for the interaction of a rail–track–bridge system was proposed. A fastener spring between the rail and the track and a friction spring between track and bridge were defined. An analysis on the track–bridge interaction was consistently performed based on a sequential analysis scheme. In particular, the following conclusions have been deduced by faithfully analyzing the longitudinal behavior when only frictional resistance exists before anchors are installed.

The frictional longitudinal resistance can be replicated by a bilinear spring similar to the longitudinal resistance of a fastener. Its maximum resistance is determined by the friction coefficient between the track slab and the bridge deck.

A separate analysis scheme overestimates the maximum resistance because the longitudinal resistances for the temperature and train load are simply summed. Accordingly, the possibility of a slip occurring due to the train load is underestimated. The QHT without anchors has only the frictional resistance between the track and bridge and can slip over the bridge in the longitudinal direction. Therefore, applying a sequential analysis scheme where temperature and train loads are considered sequentially is appropriate.

The additional rail stress can adversely increase due to the relative displacement in the track segment when train traction/braking load is applied. The friction coefficient under the QHT should be maintained at 0.3 or higher before anchor installation. When the frictional coefficient may become excessively small, low-speed operation should be guided so as to not induce traction/braking loads.

A relative slip between the track and bridge due to the temperature load occurs in all cases regardless of the friction coefficient because it is only supported by the frictional resistance from the self-weight of the track slab. In contrast, a relative slip due to train traction load occurs at the critical friction coefficient of 0.3 or below.

The maximum slip due to the temperature load was less than 10 mm. The final relative displacement will be less than this maximum value, as the temperature is repeatedly altered on a daily basis. The secondary slip due to the train load is smaller in size than the first slip due to the temperature load; however, the secondary slip may accumulate if trains are operated, as it continuously works in the same direction. When the friction coefficient is 0.5 or higher, only the first slip due to the temperature load occurs without the secondary slip due to the train load; hence, the longitudinal displacement from the train operation does not accumulate. When the friction coefficient is 0.3 or less, the secondary slip occurs from the train operation, which eventually accumulates if trains are operated repeatedly, thereby making it difficult to control the longitudinal displacement.

As constantly maintaining the friction coefficient at actual sites is difficult, anchors should be installed promptly after constructing QHTs to fix on the bridge. In this study, the proposed sequential analysis in which the frictional resistance at the underside of the track segment is taken into consideration can be useful for determining the critical friction coefficient and the appropriate longitudinal resistance of a rail fastener, as well as for verifying track safety.

**Author Contributions:** Formal analysis and writing, S.C.; methodology and writing, K.-C.L.; validation, S.Y.J.; project administration, I.L.; supervision and investigation, W.C. All authors have read and agreed to the published version of the manuscript.

**Funding:** This research was funded by the R&D Program of The Korea Railroad Research Institute, grant number PK2004B1.

**Conflicts of Interest:** The authors declare no conflict of interest.

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
