# Peer review of "Sequential Track–Bridge Interaction Analysis of Quick-Hardening Track on Bridge Considering Interlayer Friction"

_applsci, doi:10.3390/app10155066_

Round 1
Reviewer 1 Report
This paper develops a rail-track-bridge system to analyse the effect of the interlayer friction on the slip subjected to the moving train load and temperature. Generally, this is a good attempt to include quantify the effect of friction. However, the reviewer finds some issues which should be properly tackled before the publication can be recommended.
In the first paragraph of Introduction, it is suggested to do the literature review from an aboard view regarding the irregularities existing in the railway system, such as [*].
[*] Y. Song, Z. Liu, A. Rxnnquist, P. Navik and Z. Liu, "Contact Wire Irregularity Stochastics and Effect on High-speed Railway Pantograph-Catenary Interactions," in IEEE Transactions on Instrumentation and Measurement, doi: 10.1109/TIM.2020.2987457.
The reviewer’s biggest concern is the reliability of the tack-bridge model constructed in Section 2. Can the author provide any convincing pieces of evidence (such as the measurement data or comparison with exiting literature) to show the validation of the present model?
In line 125, please specify the type of beam element. Is it a Timoshenko beam or Euler beam? If it is Euler beam, please clarify the reasonability of the simplification.
In Section 3, it is not clear how the train load is applied to the model. The contact force between the rail and wheel should be time-varying and stochastic according to [#]. Please specify how the train is modelled or how the time-varying train load is described.
[#] Z. Wang, Y. Song, Z. Yin, R. Wang and W. Zhang, "Random Response Analysis of Axle-Box Bearing of a High-Speed Train Excited by Crosswinds and Track Irregularities," in IEEE Transactions on Vehicular Technology, vol. 68, no. 11, pp. 10607-10617, Nov. 2019, doi: 10.1109/TVT.2019.2943376.
Please reformat the Equations (1-4), as they are not in the right form.
The author analyses the effect of the friction coefficient on the slip. What is the normal value of the friction coefficient for the analysis object in the industry?
Another issue is that the author only considers the vertical coupling of bridge-track. The spatial coupling of bridge-track is also affected by the interlayer friction. Please comment on this limitation?
Reviewer 2 Report
Dear authors, thank you for your interesting paper. Small comments are in the attached file.
Did you investigate the influence on the bridge structure and increase of internal forces in bridge superstructure?

Round 2
Reviewer 1 Report
I accept the authors' response to my comments, and recommend the publication of this work.